# Analysis of Costs Associated with the Use of Personalized Automated Dosing Systems versus Manual Preparation in a Residential Center for the Elderly in Extremadura

**DOI:** 10.3390/healthcare11040620

**Published:** 2023-02-19

**Authors:** Mᵃ del Carmen Lozano-Estevan, Liliana Guadalupe González-Rodríguez, Rafael Lozano-Fernández, Jorge Velázquez-Saornil, José Luis Sánchez-Manzano, Iván Herrera-Peco, José Antonio Guerra-Guirao, Pilar Leal-Carbajo

**Affiliations:** 1VALORNUT Research Group, Department of Nutrition and Food Science, Faculty of Pharmacy, Complutense University of Madrid, 28040 Madrid, Spain; 2Departamento de Química en Ciencias Farmacéuticas, Facultad de Farmacia, Universidad Complutense de Madrid, 28040 Madrid, Spain; 3NEUMUSK Research Group, Departamento de Fisioterapia, Facultad de Ciencias de la Salud, Universidad Católica de Ávila, 05005 Ávila, Spain; 4Farmacia Comunitaria de Cáceres, 10001 Cáceres, Spain; 5Faculty of Health Sciences, Alfonso X el Sabio University, Avda. Universidad, 1, Villanueva de la Cañada, 28691 Madrid, Spain; 6Departamento de Farmacología, Farmacognosia y Botánica, Facultad de Farmacia, Universidad Complutense de Madrid, 28040 Madrid, Spain; 7Servicio de Farmacia, Centro de Salud del Servicio Extremeño de Salud La Roca de la Sierra, 06070 Badajoz, Spain

**Keywords:** administration, dosage, polypharmacy, cost-analysis

## Abstract

Introduction: During the SARS-CoV-2 pandemic, there has been a decrease in the supervision of the medication of subjects with chronic diseases. Customized automated dosing systems (SPDA) are devices that allow medication to be dispensed and administered, which have proven to be safe and effective for the patient and cost-effective for the healthcare system. Methods: an intervention study was carried out on patients from January to December 2019 in a residential centre for the elderly with more than 100 beds. The economic costs derived from manual dosing were compared with those of an automated preparation (Robotik Technology^®^). Results: Of the 198 patients included, 195 (97.47%) of them were polymedicated. Of the total of 276 active substances of registered medicinal products, it was possible to include them in the process of automating the preparation of the SPDA 105 active pharmaceutical ingredients. A cost reduction of EUR 5062.39 per year was found using SPDA. Taking into account the active ingredients of emblistable and non-emblistable medicines, the use of SPDA resulted in savings of EUR 6120.40 per year. The system contributed to the detection of cases of therapeutic duplication and reduced the time to prepare the medication. Conclusions: the use of SPDA is a useful and economically profitable strategy for its use in residential centres for the elderly.

## 1. Introduction

In the SARS-CoV-2 pandemic situation, the need to improve the continuity of care and ensure access to medicines in a safe and efficient manner became evident [1,2,3,4]. The pandemic has greatly affected the care of chronic pathologies [5,6]. In the case of polymedicated, non-institutionalised patients with multiple pathologies, there is also a decrease in the supervision of medication by relatives or direct caregivers, as visits have been reduced due to periods of restricted mobility [5,7].

The model of personalized, integrated pharmaceutical care becomes more relevant in this pandemic context [8,9,10], giving greater importance to the role of the community pharmacist after the act of dispensing medicines, participating in the administration of medicines with tools such as personalised dosage systems (PDS) that allow the medication to be organised using special blister packs, on set days and at set times. SPDs are an aid to pharmaceutical care integrated into the continuity of care for polymedicated patients while ensuring better adherence to the pharmacological treatment of these polymedicated patients and, therefore, saving the health system money by avoiding therapeutic non-compliance and admissions due to medication errors [11,12,13,14].

The patient receives his or her medication organised in a personalised system, which improves adherence to treatment, thereby optimising the control of consumption as it allows for the safekeeping of the remaining medication not dispensed at the pharmacy [15]. At the same time, it helps polymedicated patients to maintain their independence and safety and improves the quality of care and the efficiency and sustainability of the healthcare system [16,17,18].

In a residential care home for the elderly, patients have multiple pathologies and the average consumption of medicines is higher than for those who continue to live at home [18]. The manual administration of these medications is a common practice in residential centres without the intervention of the community pharmacist, and this practice has repercussions on patient safety because it is associated with dosage errors, non-compliance with dosage, incorrect interpretation of medical indications and lack of organisation of inventories [9]. In response to this, the use of personalised dosing systems (PDS) performed by community pharmacists in the Pharmacy Office has been proposed. Personalised dispensing systems can be prepared by the pharmacist manually or automatically by a blister robot, and in this case, we are talking about personalised dispensing automation systems (PDA). These robotic blistering systems constitute a new technology capable of reducing the time required for the preparation and distribution of medicines in individualised doses.

In general, only solid oral preparations with good physical, chemical and pharmaceutical stability are suitable for inclusion in SPDs, provided that they are stable outside the original primary packaging at room temperature for the period of unblistering, preparation, delivery and use. Non-oral dosage forms, such as ova or suppositories, and non-solid oral dosage forms, such as sachets, should be excluded from SPD. Thermolabile (refrigerator storage between 2 and 8 °C) medicinal products should also be excluded [18].

Extremadura has a population that is older and more dependent than the national average and usually resides in rural areas [19]. This has been associated with the implementation of both public and private resources to improve the care of the adult population residing in residential centres or day centres [20]. In order to overcome deficiencies in the supply and dispensation of medicines, the use of SPDA in these facilities has been proposed [21,22]. However, it is not known to what extent these systems are associated with a reduction in financial expenditure. For this reason, the main objective of this study is to compare the costs of manual drug administration with SPDA. Performed by pharmacists in a Pharmacy Office in a residential center for the elderly in Extremadura.

## 2. Materials and Methods

An intervention study was carried out in which a model of pharmaceutical care integrated into the care model of the resident population of the Ciudad Jardín Residential Centre in Cáceres was designed as the object of study, using personalised automated dosage systems (SPDA).

This study was carried out from January to December 2018 and January to December 2019 on a closed multi-pathological and elderly population in the Ciudad Jardín residential centre, with a private management model and a mixed public–private medication supply system of the Extremadura Health Service/Pharmacy Office. As it is a residential centre of more than 100 beds, part of the supply of medicines for patients is provided by the Hospital Pharmacy of the referral hospital and the medicines that the hospital does not have in stock are supplied by a Pharmacy Office.

This residential centre places fortnightly orders for medicines from the Hospital Pharmacy of the Hospital San Pedro de Alcántara according to the patient’s treatment schedule for 15 days. Residential centre does not have any medicine management programme to order the exact consumption of medicines to be consumed in the next 15 days, so the nurses make an estimate according to the existing stock and with reference to previous orders accumulating in the residential centre to create a large stock of medicines. The Ciudad Jardín Residential Centre has 240 beds. The two groups of patients, both in 2018 and 2019, were made up of the same patients in order to avoid selection bias.

During 2018, the Garden City Residential Centre used the manual pill dispenser method by the staff. Starting in 2019, throughout the year, the production of SPDA blister packs using the medicine blistering robot (RobotikTechnology^®^) was introduced at the Pilar Leal Pharmacy in Alcuéscar, Spain.

Pharmaceutical dosage forms that can be included in both manual and automated personalised drug delivery systems include only solid oral medicinal products with good physical, chemical and pharmaceutical stability, provided that they are stable outside the original primary packaging at room temperature for the period covering unblistering, preparation, delivery and use. Non-oral dosage forms, such as ova or suppositories, and non-solid oral dosage forms, such as so-bres, should be excluded from SPD. Thermolabile (refrigerator storage between 2 and 8 °C) medicinal products should also be excluded.

This modification in the way medication is administered was used to compare the costs of manual and automated production (year 2018 compared to 2019). The medication supply system was the same in both 2018 and 2019. The Garden City Residential Centre has a supply of public and private medication. Public management corresponds to the Extremadura Health System, and the private management corresponds to the Pharmacy Office.

In order to implement the model, project management techniques, process management, training for the people involved in the process and a study to assess the previous situation in terms of financial and patient safety were used in the different phases of the project’s implementation.

A homogeneous population was considered in terms of the main variables: age over 64 years of age and without a direct reference person for medication management. The following variables were chosen as study variables: age, gender, co-payment of the individual health card (IHC), pharmaceutical expenditure on medicines, medical devices and medicines not financed supplied by the Extremadura Health Service and patient co-payment at the Alcuéscar Pharmacy Office. The number of medicines consumed by each patient was also taken into account, and polymedication was considered to be the consumption of three or more medicines/per day.

The information was obtained from the electronic and paper prescriptions from the geriatric centre’s doctor, as well as from the orders for medicines received at the Residential Centre from the Hospital Pharmacy of the San Pedro de Alcántara Hospital in Cáceres. Data on the consumption of medicines were also obtained from the management software “Farmadosis” AMCO+ (FARMADOSIS S.L., Palma de Mallorca, Spain), a management programme used by the Pharmacy Office for the weekly production of SPDA medication that connects the SPDA robot with healthcare professionals of the residential centre, as well as with the technology and information systems necessary to coordinate the pharmaceutical care with healthcare professionals in the nursing of this centre. The Pharmacy Office carries out the weekly emblisting of patients’ medication for production with the SPDA system, which makes it possible to obtain metrics related to medication consumption.

All patients of this centre who agreed to participate were included in a consecutive selection process. For this purpose, the patients were given an informed consent form explaining the study in detail and that the data obtained would be anonymous and would only be used for the purpose of the study.

The inclusion criteria were as follows:-Persons with a stay of 30 days or more at the Ciudad Jardín Residence.-A user of 1 or more drugs for a period greater than or equal to 30 consecutive days of chronic medication.-Absence of a direct primary caregiver.-Sign the authorisation and informed consent included in the standard work protocol (SOP) of the SPDA of the Pharmacy Office in order to relocate their medication to the SPDA’s list at the Alcuéscar Pharmacy Office.-To have their treatments financed by the Extremadura Health System (SES), with TSI 001, 002, 003, 004 and 005, as established by Royal Decree-Law 16/2012, of 20 April, on urgent measures to guarantee the sustainability of the National Health System and improve the quality and safety of its services [23].

The exclusion criteria were as follows:-Stay of less than 30 days (either due to change of residence or death before 30 days).-Non-user of medicines.-Not signing the informed consent form.-Belonging to companies or mutual insurance companies

Discontinued medications were excluded for patients with chronic medications.

The model was subject to continuous quality assessment, with measures adapted to the needs and lines identified as possible improvements.

In September 2018, the situation analysis phase was carried out prior to the implementation of the pharmaceutical care model in the residential centre. Meetings were held with healthcare staff, especially for reviewing the processes related to patient treatment. A working methodology adapted to what they were doing was proposed so that the implementation of personalised pharmaceutical care could be integrated into the usual working system. The healthcare staff of the residential centre, responsible for administering medicines to patients, were trained in SPD systems for medicines to ensure patient safety and without extensive modification of routine clinical practice.

We retrieved data on medications consumed by patients throughout 2018 and demographic data on all patients discharged from the centre. They were segmented by age, gender and type of health coverage. Training of healthcare staff was intensified for the implementation of this new model.

To this end, the technology and information systems in place were used to coordinate pharmaceutical care with the healthcare professionals at the residential centre.

During the period analysed, in 2018 and 2019, a total of 198 patients who met the inclusion criteria for the study were included each year.

The expenditure associated with the manual dosing of the residential center in manual devices was quantified and compared with the expenditure associated with the elaboration of the SPDA made by the Pilar Leal Carbajo Community Pharmacy Office.

The efficiency of personnel and material resources was analysed. At all times, the preparation of active ingredients of medicines was ensured through the SPDA. Qualified and duly certified staff came from the community Pharmacy Office of Pilar Leal Carbajo.

The time spent by healthcare staff at the residential centre who prepared the individual pill dispensers with medicines using the manual method in 2018 was analysed, along with the associated economic expenditure, which was compared with the preparation of medication using the SPDAs in the Pharmacy Office in 2019 and its associated economic expenditure.

## 3. Results

Patients included in the study who met the inclusion criteria were 198 patients in each study year.

### 3.1. Results Regarding Polymedication:

The proportion of polymedicated patients in the study population was 97.47% of the total number of patients with any medication. When analysing the polypharmacy of the population, it was observed that only five patients had less than three drugs.

A total of 276 active pharmaceutical ingredients consumed each year by the patients included in the study were recorded. Of these, 99 active pharmaceutical ingredients were found that could not be reblistered due to their pharmaceutical form in the SPDA and were administered manually by the healthcare staff at the healthcare centre. The remaining 177 active ingredients of medicines consumed each year by the patients included in the study could be reblistered in the SPDA robot prepared in the Pharmacy Office. Of these 177, 72 active ingredients of medicines were supplied by the community Pharmacy Office through medical prescriptions, and the remaining 105 active ingredients of medicines were supplied by the Extremadura Health Service to the Ciudad Jardín residential centre.

### 3.2. Results Relating to the Expenditure Obtained for the Medication Administered by SPDA:

When analysing the data on essential medicines provided by San Pedro de Alcántara Hospital to the residential centre, a decrease in the number of units of medicines requested from the hospital by means of fortnightly orders by the residential centre was observed in 2019 compared to 2018.

The number of units of medicines provided by San Pedro de Alcántara Hospital and spent in 2019 when using the SPDA is 330,271 U (with an associated expenditure of EUR 83,374.90) compared to 2018 which resulted in 381,740 U (with an associated expenditure of EUR 88,437.29). This reduction in expenditure using the system proposed in this study was EUR 5062.39 per year.

The overall consumption of medicines (essential and non-essential) was analysed using the integrated pharmaceutical care system with SPDA: it was observed that in the model presented, a total of 452,997 units of medicines were consumed in 2018 at the cost of EUR 178,745.74. This compares to 2019, when 402,638 units of medicines were consumed with a PVP ii cost of EUR 172,625.34.

This represents a saving of EUR 6120.40, but this model has not had a significant influence on the decrease in the number of units of non-emblistable medicines.

This study did not quantify the medication changes that occurred from one year to the next (2018 to 2019) in patients, so this decrease can be partly can be associated with this fact. What was reduced was the overstocking of medicines in the residential centre by ordering medicines without precision in quantity, avoiding future expiry of these medicines.

The proposed system contributed to the detection of two cases of therapeutic duplicity, i.e., the presence of two medicines of the same therapeutic class, one of which was duplicity of analgesics and one was duplicity of analgesics; the patient had been prescribed paracetamol one gram and the combination tramadol/paracetamol 37.5 mg/325 mg, another was duplicity of anti-ulcer drugs, the patient had omeprazole 40 mg and esomeprazole 20 mg. An alert record sheet was created for the nursing staff, who systematically checked the patient’s visual check by them prior to the administration of medication, and reported the subjective perception of more cases where they self-corrected the duplicity error but did not record it prior to the implementation of the integrated pharmaceutical care model.

The study population comprised 46 men and 152 women. Nine patients were younger than 65 years, 40 patients between 65 and 80 years and 149 patients older than 80 years.

According to the classification established by the Ministry of Health, with regard to the TSI assigned, we see that 186 residents out of the 198 total have a TSI002, 93.93% of the total.

After analysing all the data according to the TSI of the patients in the residential centre, the only TSI to be considered is TSI001, TSI002 and TSI003, as no results were obtained for the rest as there were no patients in this range. From TSI001, a total of eight users were obtained for consideration. TSI002 yielded 186 users, of which 4 had a co-payment ceiling of 18.52 euros, leaving 182 residents with a co-payment ceiling of 8.23 euros for medicines. Finally, from TSI003, there was only one user who paid 40% of the cost of his medication. The data on the medicines financed and the co-payments made by patients with TSI002 were included, as well as the co-payments that patients would make if the supply were managed entirely by the pharmacy that serves the geriatric centre, instead of being shared with the Extremadura Health Service.

In the Ciudad Jardín Residential Centre, there is a shared supply between the Extremadura Health Service and the Pharmacy Office of Alcuéscar. It was observed that the pharmaceutical co-payment expenditure was lower than that which each resident should have to pay for the prescribed treatments. This is usually the case in residential centres in Extremadura with less than 100 beds; it is the user who finances this expense and not the Extremadura Health Service, according to the agreement between the Pharmaceutical Association of Cáceres and the Extremadura Health Service [3].

On analysing the consumption generated by the patients of the Ciudad Jardín Residential Centre, a deviation of pharmaceutical products non-financed pharmaceutical products supplied from the Pharmacy Department of the San Pedro de Alcántara Hospital in Cáceres. This justifies a separate quantitative analysis of the amount at PVL ii and PVP ii according to Botplus as of 2018 [24] and the number of units served during the year. These medicines are financed for patients who are in facilities with more than 100 beds but not for patients residing in their own homes or in facilities with less than 100 beds, as currently established.

In other words, in order to dispense these medicines, the Extremadura Health Service buys them at the LMP, so an expenditure of EUR 18,888.11 is imputed. However, if these same patients were to reside in their own homes or in a residential centre with fewer than 100 beds, this cost would not be borne by the Extremadura Health Service but would be financed by the patient themself, regardless of the patient’s other socio-economic variables.

This agreement creates a situation in which residents in centres with more than 100 beds would have a saving of EUR 28,119.46 and the Extremadura health service would have an expense of EUR 18,888.11.

Individual medication preparation time was accounted for both with the SPDA (2019) and manual preparation of weekly pill dispensers (2018). Both processes were carried out with two pharmacists for SPDA preparation and three nurses for the manual preparation of pill dispensers, who were familiar with each of the preparation processes. Average preparation time and human resource expenditure were calculated. The analysed expenditure for the manual preparation of weekly medication for 240 residents is shown in Table 1 and Table 2.

The cost of producing SPDA blister packs of medicines was also analysed. The costs of consumables are borne by the Pharmacy Office, which was cost-efficient for the health system, as well as all costs of supplies, personnel and legislative requirements. The expenditure analysed for automated SPDA preparation of weekly medication for 240 patients of the residential centre is shown in Table 3.

The SPDA was faster in the preparation of a patient’s monthly medication than the manual procedure that has been used to the current time. On the basis of these working times, the total cost was estimated: a total cost of EUR 3348 per medication preparation process per month for 240 patients in 2018 (manual system) and a total expenditure in 2019 (with SPDA) of EUR 1437.6 for 240 patients.

At the time of the study, as the SES supplies the user without strict control, it receives EUR 5978.30 less and has an expenditure of EUR 18,888.11. In the study, the SES has a total loss of EUR 24,866.41 (EUR 5978.30 + EUR 18,888.11). In addition to showing a difference in access to medication between residents in centres with more than 100 or less than 100 beds or at home, regardless of the socio-economic situation of each patient.

## 4. Discussion

The implementation of this model of personalised pharmaceutical care integrated into the care of institutionalised patients in the population studied shows an economic impact on both the Extremadura Health Service and the Ciudad Jardín Residential Centre. As it is included in comprehensive pharmaceutical care, with appropriate information systems, the model presented allows for the detection of duplicity and other medication errors and solving them in real-time [11,24,25]. This means improving the quality of patient prescribing and, therefore, patient compliance, contributing to the optimal control of chronic diseases in terms of medication management [26]. The professionalisation of medication supply and the presence of a pharmacist to review and monitor treatments, duplications and interactions on a personalised basis for each patient has been shown to contribute to the rationalisation of healthcare expenditure.

Non-financed medicines were quantified at PVL ii because this is the purchase price of these products, according to Botplus, as of 2018 [27]. This potentially represents a direct saving for the Extremadura Health Service in pharmaceutical expenditure for these patients in one year of using this personalised pharmaceutical care model [18]. As a cost-saving measure for the Extremadura Health System, by ceasing to request fortnightly bulk orders, non-financed medicines and medical devices would change the financer by being to be charged to the patient in centres with more than 100 beds [17]. Until now, in facilities with less than 100 beds or residents at home, this expenditure is charged to the user.

The residential centre that was under this system would have more hours per month of qualified health personnel to reassign to other occupations that would improve the quality of the centre and its patients, saving time in storage and placement of medicines and checking expiry dates, as well as saving money on the purchase of medication trolleys and their maintenance.

The pharmacy that participates in the pharmaceutical care of this model, which was implemented in the Ciudad Jardín residential centre, assumed direct investment costs in technology and information systems to carry out the SPDA of medicines, consumable materials necessary for weekly production, and the relevant legal obligations. Likewise, the hiring of personnel to provide a quality service to the residential centres is financially compensated by the profit margin on the increase in sales of the medicines and medical devices discussed during this study.

It is therefore important to consider this model of integrated pharmaceutical care in residential centres as an alternative to the current model in order to guarantee the sustainability of social and healthcare services while at the same time providing quality, patient safety and equity.

A more extensive study on equity of access to medication for patients living in facilities with less than 100 beds and no and not institutionalized would be needed [28].

One of the limitations of the study is that it focuses on a single centre and takes as a reference the medication control system of the residents of a single centre. However, it would be useful to extrapolate this system to other centres, so we believe that future research can be based on our work in order to contribute more scientific quality to the community. To date, there is no article that discusses this process of cost analysis associated with personalised dosing systems, comparing the two possibilities: manual versus automated preparation.

## 5. Conclusions

SPDAs are effective tools to optimise the pharmacological treatment of institutionalised elderly people and are safer and less costly than manual dosing of medicines. Their implementation in facilities has a positive economic impact on health services, as well as a way of incorporating integrated pharmaceutical care.

## Figures and Tables

**Table 1 healthcare-11-00620-t001:** Patients included in the study with polypharmacy.

Number of Medicines	Patients *(N = 198)*
1	1
2	4
3	8
4	22
5	25
6	26
7	34
8	26
9	23
10	8
11	9
12	8
13	1
14	3
Total patients	198

**Table 2 healthcare-11-00620-t002:** Economic study for the manual preparation of weekly pill dispensers for 240 residents of the Ciudad Jardín centre.

Manual preparation time of 1 Pillbox per week with 4 intakes per day	14–15 min/pill dispenser
Number of pill dispensers per month	4.33 units (average for 52 weeks per year)
Total preparation time for one month for one patient	64.95 min/month/patient 1.08 h/month/patient
Cost of salary + social insurance of nursing personnel	12.18 EUR/h (according to agreement)
Cost of manual pill dispenser preparation per patient	1.08 h/month/patient × 12.18 EUR/h = 13.15 EUR/month/patient Miscellaneous maintenance costs: 0.80 EUR/patient/month
Costs of consumables and supplies to be borne by the Geriatric Centre	Pill dispenser renewal: 35 EUR/unit (10% annual renewal) Annual medication trolleys: 1.500 EUR/unit (one unit every 3 years) Cleaning products, gloves: 30 EUR/month
Costs of consumables and supplies to be borne by the Geriatric Centre	13.95/month/patient × 240 patients = 3348 EUR/month for 240 patients

**Table 3 healthcare-11-00620-t003:** Economic study for the automated preparation of weekly SPDA for 240 patients of the residential centre Ciudad Jardín.

Manual preparation time of 1 SPDA per week with 4 intakes per day	1 min/patient/28 sachets
Number of SPDA per month	120 sachets (average for the 52 weeks with medication changes during the month)
Total preparation time for one month for one patient	1 min/month/patient 0.066 h/month/patient
Cost of salary + S.S. of pharmaceutical personnel	16.83 EUR/h (according to agreement)
SPDA preparation cost per patient	0.066 h/month/patient × 16.83 EUR/hour = 1.11 EUR/month/patient 0.03 EUR/bag × 120 bags/month/patient/patient = 3.60 EUR/month/patient Other expenses attributable to the O.F. = 1.28 EUR/month/patient
Costs of consumables and supplies to be borne by the F.O.	Ink ribbon and pouch film: 120 EUR/4.000 pouches produced (0.03 EUR/bag)
	Technology investments and monthly software maintenance: 150 EUR/month
	Electricity, insurance and equipment: 30 EUR/month
Total SPDA preparation for 240 patients	5.99/month/patient × 240 patients = 1437.60 EUR/month for 240 patients assumed by the Pharmacy Office of Alcuéscar

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
