# Peer review of "Analysis of Costs Associated with the Use of Personalized Automated Dosing Systems versus Manual Preparation in a Residential Center for the Elderly in Extremadura"

_healthcare, 2023, doi:10.3390/healthcare11040620_

Round 1

Reviewer 1 Report

The authors of the manuscript titled "Analysis of costs associated with the use of personalized automated dosing systems versus manual preparation in a residential center in Extremadura." have tried to evaluate the cost effectivity of automated pill dosing system in an adult care setup.  The study heavily lacks relevant details and the presentation of it has made it nearly incomprehensible. The introduction needs better references, a compelling story as to why this study is needed. The methods section needs major overhauling to include details in a more cohesive manner and the result section needs clarity as to what was observed. There are several sentences that has repeating words that do not make any sense, as well as the monetary amounts are not readily understandable.

Author Response

Dear reviewer:

A thorough revision of each section has been carried out, making major modifications in each one of them, with the aim of making all the sections more comprehensible and presenting the data of the study in a more coherent and clearer way. In addition, the written language has been revised and the monetary amounts are represented according to the journal's guidelines. Thank you very much for your contributions to increase the quality and evidence of our manuscript.
Best regards.

Reviewer 2 Report

This is an interesting paper regarding the analysis of costs associated with the use of personalized automated dispensing systems versus manual preparation in a residential center in the Extremadura region.

The introduction provides relevant information and the method is clear. The results were well presented and discussed, leading to the conclusion that customized automated dosing systems (SPDAs) represent an effective tool for optimizing the pharmacological treatment of institutionalized elderly people, as well as being safer and cheaper than manual drug dosing. Furthermore, the authors explained the limitations of the study, since it focuses on one center and that it would be useful to extrapolate this system to other centers as well.

I have no further concerns from the pharmacological point of view.

Author Response

Dear reviewer:

Thank you very much for your considerations. Best regards.

Reviewer 3 Report

The present manuscript, “Analysis of costs associated with the use of personalized automated dosing systems versus manual preparation in a residential center in Extremadura” discussed the cost-effectiveness of personalized manual and robotic treatment. The period of SARS-CoV-2 pandemic leaves patients with chronic diseases in negligence. Further, the cost of manual medication was subjected to a huge economical burden. However, self-medication is associated with the serious error of dosing. Thus, the proposed study of automated medication and study of its cost-effectiveness was essential.

The background of the study was well established at the starting of the manuscript. Further, the result obtained was systematically elaborated in the result section. The number of subjects allotted for the study was sufficient to draw a valid conclusion. The present manuscript is accepted in its present form.       

Author Response

(The authors gave the same response as above.)

Reviewer 4 Report

The topic is good and will contribute to the research and beneficial to the scientific community. However, it would be better to consider the below comments for improvements:

1.    Title:

Why the word “geriatric” not included in title as, the use of SPDA is a useful and economically profitable for its use in geriatric residences.

2.    Abstract:

The abstract of 201 words is provided – it would be better if abstracts reflect the whole process/methods, objective and inclusion & exclusion criteria of the present study. The current layout is not truly reflecting the present study.

3.    Introduction:

You many need to mention very clearly the current gaps, what is the rationale for doing this present and how it is helpful in pandemic/COVID 19 situation.

Needs to revise the sentence “SPDs allow for integrated pharmaceutical integrated into the continuum of care”.

The words “Drug” and “Medicine” used throughout the article; you can standardize to use suitable term wherever possible.

4.    Materials and Methods:

You can add on the information that, what is the indication of medications those dispended by SPDA? Also, is there validation done by the expert team for the data extracted and for its appropriateness?  

Is there any criteria for selection of medication for this study?

5.    Results:

You can add on the information related to the – medication mostly dispense, indication and problems encounter if any while dispensing by SPDA, if any.

6.    Discussion:

It would be better to discuss the points relevant to the future prospectus/direction of SPDA use.

7.    Conclusion:

Is clear and understandable.

8.    Limitation: Is all the medication/formulations can be dispensed by SPDA? Also, those medication needs special instructions/counselling on how to use/precautions needs to take?

9.    References:

Please check and double confirm the references cited according to journal author instructions. 

Author Response

Dear Reviewer 4:
The topic is good and will contribute to the research and beneficial to the scientific community. However, it would be better to consider the below comments for improvements: 1. Title: Why the word “geriatric” not included in title as, the use of SPDA is a useful and economically profitable for its use in geriatric residences. Thank you very much. The comment has been taken into account and the term has been replaced, you can check it in the title of the manuscript. In addition, we have unified the whole text to the use of the terminology residential care home for the elderly, thank you. 2. Abstract: The abstract of 201 words is provided – it would be better if abstracts reflect the whole process/methods, objective and inclusion & exclusion criteria of the present study. The current layout is not truly reflecting the present study. More data from the study has been included in the summary to better reflect the whole process, thanks for the appreciation.
3. Introduction: You many need to mention very clearly the current gaps, what is the rationale for doing this present and how it is helpful in pandemic/COVID 19 situation. Justifications have been added in the introduction to the study, thanks. Needs to revise the sentence “SPDs allow for integrated pharmaceutical integrated into the continuum of care”. This sentence has been modified on lines 63-67: "SPDs are a tool to assist in the pharmaceutical care integrated in the continuity of care for polymedicated patients, while guaranteeing greater therapeutic adherence to the pharmacological treatment of these polymedicated patients and therefore economic savings for the health system by avoiding therapeutic non-compliance and admissions due to errors in taking medication[11-14]."

Thank you very much. The words “Drug” and “Medicine” used throughout the article; you can standardize to use suitable term wherever possible. Your consideration has been taken into account and the term has been replaced throughout the text, thank you very much. 4. Materials and Methods: You can add on the information that, what is the indication of medications those dispended by SPDA? Also, is there validation done by the expert team for the data extracted and for its appropriateness? A text explaining which medicines can be included in SPDAs has been included on lines 143-150, thank you very much for the appreciation. Is there any criteria for selection of medication for this study? It does not exist, they are all the medicines that the patients included in the study are prescribed in their pharmacological treatment. Thank you. 5. Results: You can add on the information related to the – medication mostly dispense, indication and problems encounter if any while dispensing by SPDA, if any. This information has not been studied, as it is not the objective of the study. We take up this idea for future lines of research, thank you for your contribution. 6. Discussion: It would be better to discuss the points relevant to the future prospectus/direction of SPDA use. More content has been added to this discussion section. You can check it on lines 426-441, thank you very much. 7. Conclusion: Is clear and understandable. Thanks. 8. Limitation: Is all the medication/formulations can be dispensed by SPDA? Also, those medication needs special instructions/counselling on how to use/precautions needs to take? A text explaining the medicines that can be included in the SPDA has been included, thanks.

9. References: Please check and double confirm the references cited according to journal author instructions. All references have been checked and a citation has been carried out according to the journal's guidelines, thank you very much.

Please review the attached document.

Best regards

Round 2

Reviewer 1 Report

Authors' have improved the manuscript enough that it is understandable now. I have no further comments.